# Role of Clinical-Demographic Data in Survival Rates of Advanced Laryngeal Cancer

**DOI:** 10.3390/medicina57030267

**Published:** 2021-03-15

**Authors:** Eugenia Allegra, Maria Rita Bianco, Massimo Ralli, Antonio Greco, Diletta Angeletti, Marco de Vincentiis

**Affiliations:** 1Department of Health Science, University of Catanzaro, 88100 Catanzaro, Italy; mrbianco@unicz.it; 2Department of Sense Organs, Sapienza University of Rome, 00185 Rome, Italy; massimo.ralli@uniroma1.it (M.R.); antonio.greco@uniroma1.it (A.G.); diletta.angeletti@uniroma1.it (D.A.); 3Department of Oral and Maxillofacial Sciences, Sapienza University of Rome, 00185 Rome, Italy; marco.devincentiis@uniroma1.it

**Keywords:** total laryngectomy, tobacco-related cancers, laryngeal cancer, advanced laryngeal cancer

## Abstract

*Background and Objectives*: Laryngeal cancer is one of the most common cancers in the upper aerodigestive tract, and tobacco and alcohol habits are the most relevant risk factors. The role of these risk factors in the incidence of laryngeal carcinomas is well known, yet only a few studies have been conducted on their role as risk factors of prognosis. The aim of the study was to assess the impact of clinical–demographic data on overall survival (OS), disease-free survival (DFS), and disease-specific survival (DSS) in patients with advanced-stage laryngeal cancer (Stage III–IV) who underwent total laryngectomy. *Materials and Methods*: This retrospective study was carried out on patients with Stage III–IV laryngeal squamous cell carcinoma treated with total laryngectomy between 2004 and 2014. For each patient, clinical and anamnestic data were collected and collated in a database, including alcohol and smoking habits. *Results*: Considering the variable age, family history, alcohol, grading, subsite, stage, pT stage, pN stage, and adjuvant therapy, no statistical significance was found for five-year OS. Smoking was the only variable that was statistically significant (*p* = 0.0043). A relevant difference was noted in the five-year DFS between pN-negative and pN-positive tumors (74.3% vs. 55.26%, respectively; *p* = 0.056), and a statistically significant difference was found between non- and ≤20 cigarettes/day smokers and heavy smokers (77.78% vs. 53.66%, respectively; *p* = 0.021). The five-year disease-specific survival rate was 68.83%, and a significant difference was detected for the smoking and pN stage variables. Heavy smokers (43.90% died vs. 16.67% of the non- and ≤20 cigarettes/day smokers; *p* = 0.0057) and pN-positive (42.1% died vs. 20.51% of the pN-negative patients; *p* = 0.042) patients had a worse prognosis. *Conclusion*: Smoking in our study was found to be an important independent risk factor for worse OS and DSS in patients with advanced laryngeal cancer.

## 1. Introduction

Laryngeal cancer is one of the most common cancers in the upper aerodigestive tract and represents 4.5% of all malignancies [1]. Tobacco and alcohol habits are the most relevant risk factors, and the roles of gastro-esophageal reflux and human papillomavirus infection are still debated. In the last three decades, the incidence of laryngeal cancer has increased by 12%, with the highest number of deaths recorded in Europe [2]. The type of treatment depends on the stage of disease at diagnosis, and can benefit from different modalities, including transoral laser microsurgery, organ partial horizontal laryngectomy, and radiotherapy, with good oncological and functional outcomes. In recent years, the number of open preservation surgeries has increased, open partial laryngectomies have been demonstrated to be a valuable alternative to total laryngectomy for surgical salvage in laryngeal squamous cell carcinoma (LSCC) after chemoradiotherapy failure and transoral laser surgery recurrence. The purpose of this surgery is to avoid the highly mutilating intervention of a total laryngectomy [3] and to maintain laryngeal function with respect to oncological radicality [4,5]. However, to date, most patients affected by laryngeal carcinoma are diagnosed with an advanced stage of the disease, requiring total laryngectomy. These patients have a poor prognosis due to the occurrence of locoregional recurrence, distant metastasis, or second tumors. The role of some prognostic factors, such as smoking and alcohol, on the incidence of laryngeal carcinomas is well known, yet only a few studies have been conducted on the role of clinical–demographic variables as risk factors of prognosis. The aim of this study was to assess the impact of clinical–demographic data on overall survival (OS), disease-free survival (DFS), and disease-specific survival (DSS) in patients with advanced-stage laryngeal cancer (Stage III–IV) who underwent total laryngectomy.

## 2. Material and Methods

### 2.1. Patients

This retrospective study was carried out on patients with Stage III–IV laryngeal squamous cell carcinoma treated with total laryngectomy between 2004 and 2014 at the Otolaryngology Unit of the Department of Health Sciences, University of Catanzaro, Italy. All patients were informed of the benefits, risks, possible complications, and alternatives to surgery before providing their informed consent. This study was approved by the Institutional Review Boards at the University of Catanzaro and the requirement to obtain written informed consent from patients was waived due to the retrospective nature of the investigation. To protect the patients’ privacy, their personal information was appropriately anonymized prior to analysis. All included patients had been submitted to total laryngectomy and followed for a minimum of 60 months. We excluded from this study patients lost at follow-up or with incomplete anamnestic or clinical data. Patients preoperatively underwent videolaryngoscopy with a flexible endoscope and videolaryngostroboscopy. The neck was examined by palpation, whereas ultrasound, computed tomography (CT), and magnetic resonance imaging (MRI) were included for staging. The stage was determined in accordance with the 7th edition of the TNM classification established by the American Joint Committee for Cancer. All patients were followed up within 1 month of surgery, every 3 months for 3 years, and every 6 months thereafter. Follow-up visits included clinical examination, laryngoscopy, and radiological examinations, including neck ultrasound, chest X-ray every six months, and CT or MRI every year or according to clinical evidence. For each patient, clinical and anamnestic data were collected and collated in a database, including alcohol and smoking habits. To collect data on alcohol habits, the recruited subjects were divided into subjects who consumed more than 500 mL per day, and subjects who did not consume alcohol or consumed less than 500 mL per day. To collect data on smoking habits, at the time of the diagnosis, we considered heavy smokers who smoked >20 cigarettes per day, moderate ≤20 cigarettes per day, and non-smokers who had never smoked or had stopped.

### 2.2. Statistical Analysis

Statistical analysis was performed using MedCalc Software (v9.0; MedCalc Software bvba, Ghent, Belgium). For normally distributed data, the mean and standard deviations were utilised. Categorical data were reflected as counts and percentages. Pearson’s chi-squared and/or Fisher’s exact tests were used to identify differences in the demographic and clinicopathologic data between cohorts. The D’Agostino–Pearson test was used for normally distributed data. Comparisons of categorical variables among the groups of patients were performed by means of either Chi-square test or Fisher Exact test when appropriate. The five-year overall survival time, five-year disease-free survival, and five-year disease-specific survival rate were assessed. The variables considered in the survival analysis included age, T and N status, adjuvant therapy, tumor subsite, alcohol habits, and smoking habits. The Kaplan–Meyer method was used for the survival. The log-rank test was used to compare survival curves between groups. A multivariate analysis was performed using the Cox regression method to evaluate the independent contribution of the variables to overall survival, disease free survival and disease specific survival. Hazard Ratio (HR) and 95% Confidence Interval (95% CI) are shown; a *p*-value < 0.05 was considered statistically significant. OS was defined as the time interval from surgery ± adjuvant treatment until death (from any cause). DFS and DSS were defined as the time intervals from treatment until loco-regional recurrence and death due to the disease. All deaths due to other causes were considered censored.

## 3. Results

A total of 77 patients were included in the study; two were female and 75 were male. The mean age at diagnosis was 66 ± 9.7 SD years. The mean follow-up time was 76.0 ± 43.3 SD months. In total, 23 of the 77 (29.9%) patients had a family history of cancer disease, whereas 41 of the 77 (53.2%) were non-smokers or smoked ≤20 cigarettes/day, and 36 (46.8%) smoked >20 cigarettes/day. In addition, 34 of the 77 (44.1%) patients were non-drinkers or drunk less than 500 mL/day, and 43 drunk more than 500 mL/day. Furthermore, 31 of the 77 (40.2%) had tumors with a glottic location, whereas 23 (29.9%) had supraglottic tumors and the other 23 (29.9%) had transglottic tumors. The histological grading was G1 in five of the 77 (6.5%) patients, G2 in 42 (54.5%), and G3 in 30 (39.0%). All patients had free surgical margins. Moreover, 44 (57.1%) of the patients were classified as Stage III, and 33 (42.9%) were classified as Stage IV. According to the cTNM classification, 21 lesions (27.3%) were staged as T2, 35 (45.4%) as T3, and 21 (27.3%) as T4. Post-operative pT classification was pT2 in 21 (27.3%) cases, pT3 in 35 (45.4%), and pT4 in 21 (27.3%). The lymph node clinical status was classified as N0 in 39 patients, N1 in 12, N2 in 23, and N3 in three. A total of 63 of the 77 patients received neck dissection and, of those, 38 had metastatic lymph nodes. In 6 of the 38 metastatic lymph node an extranodal invasion has been observed. Furthermore, 34 of the 77 patients underwent surgery plus radiotherapy, whereas eight underwent radiotherapy plus chemotherapy (Table 1). Comorbidities were present in 51 of the 77 patients, in 23/51 were affected by hypertension, 18/51 diabetes, 10/51 hypertension and diabetes.

During the follow-up period, 14 of the 77 patients (18.2%) presented loco-regional recurrence after a mean time of 22.5 ± 11.2 SD months. Seven of these patients developed lymph node metastasis, and seven developed local recurrence. A total of 13 (16.9%) of the 77 patients developed a second tumor or distant metastasis after a mean time of 23.4 ± 12.7 months. Seven of the 77 patients (9.0%) developed a pulmonary metastasis, three esophagus cancer, two liver carcinoma, and one brain metastasis. The five-year overall survival rate was 62.3%; 29 of the 77 patients died. Specifically, 13 patients died because of distant metastasis or a second tumor, 10 died after loco-regional recurrence, and six died of non-cancer related diseases. Considering the variable age (*p* = 0.73), family history (*p* = 0.18), alcohol (*p* = 0.08), grading (*p* = 0.22), subsite (*p* = 0.50), stage (*p* = 0.33), pT stage (*p* = 0.47), pN stage (*p* = 0.29), and adjuvant therapy (*p* = 0.19), no statistical significance was found for five-year OS. Smoking was the only variable that was statistically significant (*p* = 0.0043), wherein the five-year OS rate was 77.78% for non- and ≤20 cigarettes/day smokers and 48.78% for heavy smokers (>20 cigarettes/day) (Figure 1). Cox proportional hazards regression, used to analyze the role of clinical–demographic variables as risk factors for overall survival, showed that smoking was the only independent risk factor of worse OS (Table 2).

The five-year disease-free survival was 64.94%. None of the variables, i.e., age (*p* = 0.50), family history (*p* = 0.35), alcohol (*p* = 0.08), grading (*p* = 0.41), subsite (*p* = 0.48), stage (*p* = 0.42), pT stage (*p* = 0.48), pN stage (*p* = 0.29), or adjuvant therapy (*p* = 0.39) showed a statistically significant difference. A relevant difference was noted in the five-year DFS between pN-negative and pN-positive tumors (74.3% vs. 55.26%, respectively; *p* = 0.056), and a statistically significant difference was found between non- and ≤20 cigarettes/day smokers and heavy smokers (77.78% vs. 53.66%, respectively; *p* = 0.021) (Figure 2). The Cox proportional hazards regression model was used to analyze the role of anamnestic and clinical variables as risk factors for DFS, revealing that smoking was the only independent prognostic factor for worse DFS (Table 3). The five-year disease-specific survival rate was 68.83%. A significant difference was detected for the smoking and pN stage variables. Heavy smokers (43.90% died vs. 16.67% of the non- and ≤20 cigarettes/day smokers; *p* = 0.0057) and pN-positive (42.1% died vs. 20.51 of the pN-negative patients; *p* = 0.042) patients had a worse prognosis (Figure 3). An evident difference was also found for patients with an alcohol habit. The five-year DSS was 42.83% and 21.43% for heavy drinkers and non-drinkers or moderate drinkers, respectively (*p* = 0.058). No significant difference was noted for any other variable. Cox proportional hazards regression was used to analyze the role of anamnestic and clinical variables as risk factors for DSS, revealing that, even in this case, smoking was the only independent prognostic factor of worse survival (Table 4). When analyzing the clinical–anamnestic characteristics of non- and ≤20 cigarettes/day smokers and >20 cigarettes/day smokers, a statistically significant difference was observed in the patient distribution according to follow-up time (Table 5). In the non- and ≤20 cigarettes/day smoker group, eight patients died (two from lymph node metastasis, three from distant metastasis, and three from non-cancer-related diseases), whereas in the >20 cigarettes/day smoker group, 21 patients died from loco-regional recurrence, 10 from distant metastasis or a second tumor, and four from non-cancer-related diseases).

## 4. Discussion

According to a recent study by Kulhanova et al. [6], laryngeal carcinoma represents the second most prevalent tobacco-related cancer following lung cancer. Its incidence has increased during the last three decades and Europe remains the continent with the highest incidence and mortality [2]. During the past three decades, laryngeal cancer mortality has marginally decreased in the global population, but a constant increase has been observed during the past 10 years [7,8]. To date, few studies have examined the role of risk factors on laryngeal cancer survival. Epidemiological studies report contrasting results regarding the association between cigarette smoking and alcohol consumption on survival from laryngeal cancer. In our study, we found a significant association between smoking and survival; 41.4% of patients who smoked heavily died from causes related to laryngeal cancer versus 13.8% of the patients who were non- or ≤20 cigarettes/day smokers. In 24.3% of the cases, the cause of death was due to the occurrence of distant metastases. Lung cancer was the most common distant metastasis, accounting for 53.8% of all metastatic sites. Some studies have reported an association between smoking and survival, and no correlation with alcohol consumption [9,10,11,12]. More recently, Girardi et al. [13], in a study on the association of lifestyle habits and clinical data as a prognostic factor of survival for head and neck cancer, did not find any association between smoking and laryngeal cancer survival. However, they found a correlation between alcohol consumption and overall survival. The population of this study was inhomogeneous and was composed of patients from northern Italy, Brazil, and Japan with different lifestyle habits. However, these results are close to those reported by Pan et al. [14], who found that alcohol consumption also correlated with worse DSS.

Our study, for the first time, considered a population of patients with laryngeal cancer in an advanced stage, all of whom underwent total laryngectomy. Unlike patients with laryngeal carcinoma who have undergone conservative treatments, the patients who smoked in our study had to stop smoking due to the functional outcome of the total laryngectomy. Therefore, the data from the patients in this study pertaining to smoking being an independent risk factor of worse prognosis indicates that the carcinogens contained in smoke are able to exert their carcinogenic capacity even after one has quit smoking. The role of the metabolic derivatives of smoke on the neoplastic transformation process of the upper airway’s mucosa is known. Their action is directed toward the structure of cellular and mitochondrial DNA [15], causing genetic mutations that lead to the genesis of cancer stem cells (CSSs) [16]. CSSs have the ability to remain dormant for a long time and to migrate to other organs distant from where they were generated. This would explain the greater risk, in heavy smokers, of developing locoregional relapses or distant metastases. Furthermore, tobacco derivatives induce alterations of DNA repair genes. Therefore, heavy smokers have a higher probability of mutations of these genes and, consequently, a reduced ability to repair any DNA damage, a condition that exposes the individual to a risk of relapses and metastases. [17].

Our study has some limitations, which are represented by the number of patients taken into consideration. Moreover, some other risk factors, such as HPV infection, were not considered. However, our study has strengths represented by the fact that the sample was homogeneous in terms of site and treatment modality, all patients were treated by the same surgery team, and all patients had a long follow-up period. It is essential to further encourage smoking prevention campaigns. In light of the obtained results, it is clear that it is important to stop smoking, but it is even more important to never have smoked so as to avoid the late effects of the action of tobacco carcinogens on prognosis.

## 5. Conclusions

Smoking in our study was found to be an important independent risk factor for worse OS and DSS in patients with advanced laryngeal cancer. Heavy smokers had a higher risk of death from loco-regional recurrence and distant metastasis. However, further studies are needed on a larger scale to better understand the role of smoking as a negative prognostic factor in advanced laryngeal carcinomas.

## Figures and Tables

**Figure 1 medicina-57-00267-f001:**
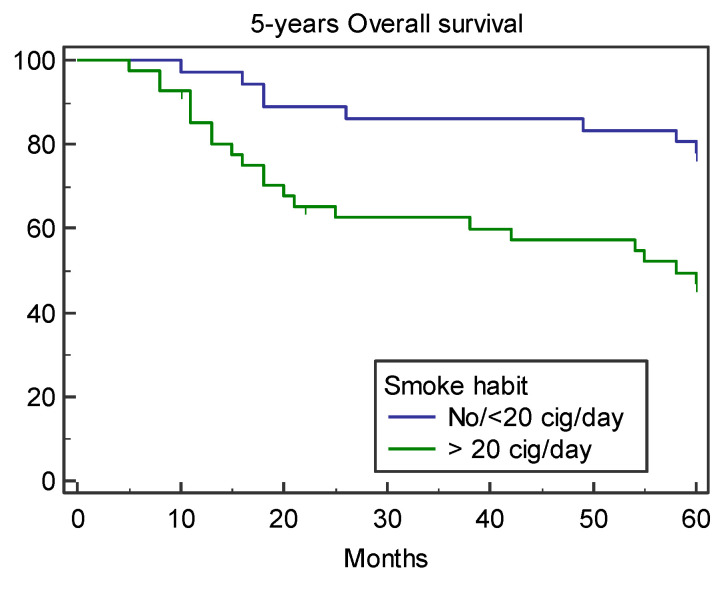
Five year overall survival according to smoke habit.

**Figure 2 medicina-57-00267-f002:**
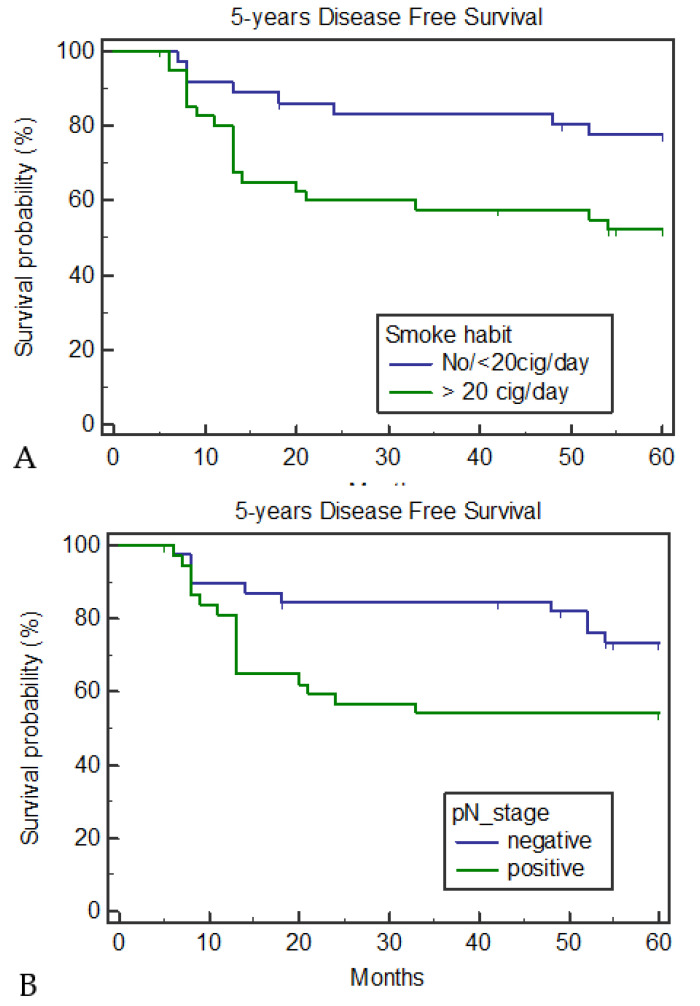
Five year disease-free survival according to smoke habit (**A**) and pN stage (**B**).

**Figure 3 medicina-57-00267-f003:**
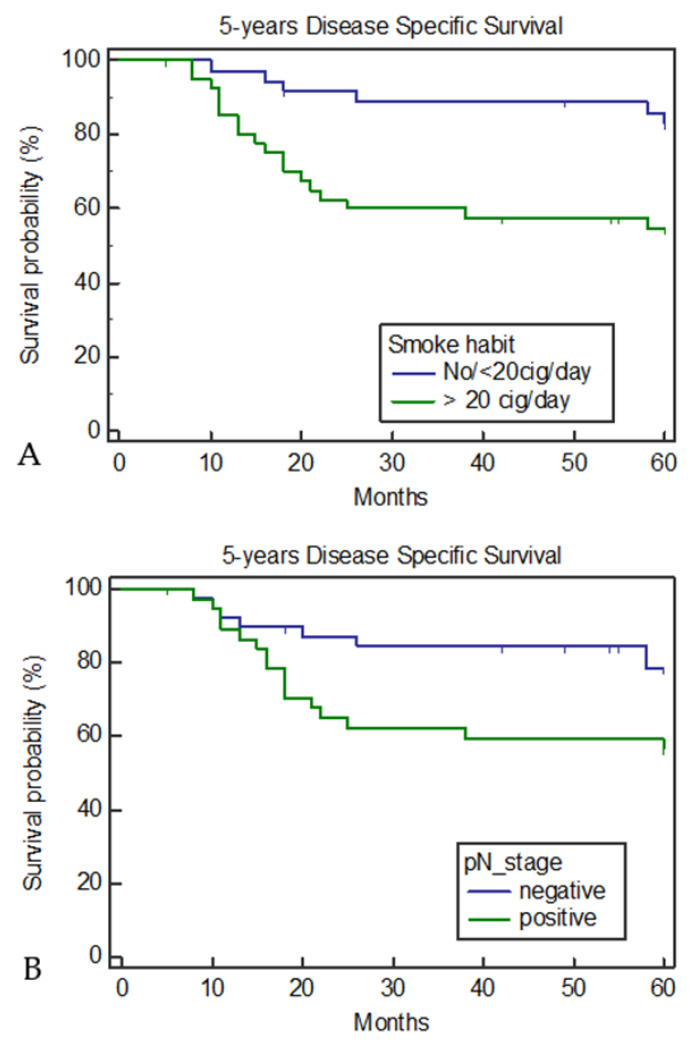
Five year disease-specific survival according to smoke habit (**A**) and pN stage (**B**).

**Table 1 medicina-57-00267-t001:** Patients clinical-demographic data.

Data	Patients n.77
**AGE**	
Mean (years) ± SD	66.0 ± 9.7 SD
**Follow-up**	
Mean ± SD (months)	76.0 ± 43.3 SD
**Family history**	
No	54
Yes	23
**Smoke habit**	
No/≤20 cig/day	36
>20 cig/day	41
**Alcohol use**	
≤500 mL/day	34
>500 mL/day	43
**Grading**	
1	5
2	42
3	30
**Subsite**	
Supraglottic	13
Transglottic	23
Glottic	31
**pT STAGE**	
2	21
3	35
4	11
**pN STAGE**	
N0	39
N+	38
**Neck Dissection**	
No	11
Yes	66
**Adjuvant therapy**	
No	39
Yes	38

Baseline cumulative hazard function.

**Table 2 medicina-57-00267-t002:** Cox proportional-hazards regression analysis of the clinical-anamnestic data as risk factors on 5-years Overall Survival.

Variable	b	SE	Wald	*p*	Exp(b)	95% CI of Exp(b)
Age	−0.13	0.44	0.09	0.75	0.87	0.36 to 2.07
Family_History	−0.35	0.53	0.43	0.50	0.70	0.24 to 2.01
Alcohol	−0.70	0.45	2.41	0.12	0.49	0.20 to 1.20
Smoke	1.29	0.44	7.32	0.00	3.33	1.39 to 7.98
Grading	0.03	0.24	0.01	0.90	1.03	0.63 to 1.68
Subsite	−0.09	0.47	0.04	0.83	0.90	0.36 to 2.28
Stage	−0.24	0.54	0.20	0.64	0.78	0.26 to 2.27
pT_stage	0.25	0.32	0.60	0.43	1.29	0.67 to 2.45
pN_stage	−0.00	0.46	0.00	0.99	0.99	0.40 to 2.47
Adjuvant_therapy	0.51	0.40	1.58	0.20	1.67	0.75 to 3.73

**Table 3 medicina-57-00267-t003:** Cox proportional-hazards regression analysis of the clinical-anamnestic data as risk factors on 5-years Disease Free Survival.

Risk Factors	b	SE	Wald	*p*	Exp(b)	95%CI of Exp(b)
Age	−0.37	0.45	0.66	0.41	0.68	0.27 to 1.69
Family_History	−0.12	0.55	0.04	0.82	0.88	0.29 to 2.63
Alcohol	−0.85	0.46	3.32	0.06	0.42	0.17 to 1.06
Smoke	0.83	0.45	3.33	0.06	2.29	0.94 to 5.62
Grading	−0.10	0.25	0.15	0.69	0.90	0.54 to 1.48
Subsite	−0.14	0.47	0.09	0.75	0.86	0.34 to 2.18
Stage	−0.43	0.54	0.64	0.42	0.64	0.22 to 1.87
pT_stage	0.21	0.32	0.43	0.51	1.23	0.65 to 2.34
pN_stage	0.54	0.47	1.31	0.25	1.71	0.68 to 4.33
Adjuvant_therapy	0.62	0.42	2.16	0.14	1.86	0.81 to 4.26

Baseline cumulative hazard function [Show].

**Table 4 medicina-57-00267-t004:** Cox proportional-hazards regression analysis of the clinical-anamnestic data as risk factors on 5-years Disease Specific Survival.

Covariate	b	SE	Wald	*p*	Exp(b)	95% CI of Exp(b)
Age	0.08	0.47	0.029	0.86	1.08	0.42 to 2.76
Family_History	−0.25	0.62	0.16	0.68	0.77	0.22 to 2.66
Alcohol	−0.67	0.47	1.97	0.15	0.50	0.19 to 1.30
Smoke	1.11	0.50	4.94	0.02	3.05	1.14 to 8.16
Grading	−0.31	0.25	1.56	0.21	0.72	0.44 to 1.19
Subsite	−0.05	0.51	0.01	0.91	0.94	0.34 to 2.60
Stage	−0.21	0.57	0.13	0.70	0.80	0.26 to 2.47
pT_stage	0.30	0.34	0.79	0.37	1.35	0.69 to 2.67
pN_stage	0.48	0.51	0.90	0.34	1.62	0.59 to 4.44
Adjuvant_therapy	0.81	0.45	3.19	0.07	2.25	0.92 to 5.50

**Table 5 medicina-57-00267-t005:** Clinical-demographic patients data and smoke habit.

Data	Smoke Habit No ≤ 20 cig/day	Smoke Habit > 20 cig/day	*p* Value
AGE			
Mean (years)	65.2 ± 9.6 SD	65.2 ± 9.4 SD	0.99
Follow-up			
mean ± SD(months)	90.9 ± 34.6 SD	62.9 ± 46.3 SD	0.003
Family history			
No	23	31	
Yes	13	10	0.32
Alcohol use			
No/<500 mL/day	18	16	
>500 mL/day	18	25	0.36
Grading			
1	2	3	
2	21	21	
3	13	17	0.07
Subsite			
Supraglottic	9	14	
Transglottic	12	11	
Glottic	15	16	0.65
pT STAGE			
2	10	11	
3	17	18	
4	9	12	0.91
pN STAGE			
N0	22	17	
N+	14	24	0.11
Neck Dissection			
No	9	5	
Yes	27	36	0.23
Adjuvant therapy			
No	16	23	
Yes	20	18	0.36

## Data Availability

Not applicable.

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
