# Peer review of "Role of Clinical-Demographic Data in Survival Rates of Advanced Laryngeal Cancer"

_medicina, 2021, doi:10.3390/medicina57030267_

Round 1

Reviewer 1 Report

The article investigates the impact of clinical–demographic data on overall survival, disease-free survival and disease-specific survival in patients with advanced-stage laryngeal cancer who underwent total laryngectomy. The manuscript was presented in an intelligible fashion and written in standard English. The weaknesses of the paper include:

Introduction

Line 38 you citated “ ….with the highest number of deaths recorded in Europe [2].” - This data remains in contradiction with WHO International Agency for Research on Cancer information

https://gco.iarc.fr/today/online-analysis table?v=2020&mode=population&mode_population=continents&population=900&populations=900&key=asr&sex=0&cancer=14&type=1&statistic=5&prevalence=0&population_group=0&ages_group%5B%5D=0&ages_group%5B%5D=17&group_cancer=1&include_nmsc=1&include_nmsc_other=1

Line 41 you stated “In recent years, the number of open preservation surgeries has increased.” - Could you please give some evidence from the literature to support this statement, because in my opinion the increase is present but in endoscopic intralaryngeal procedures not in open partial laryngectomies.

Line 50 and thorough other section of the manuscript you use the phrase “….as risk factors of prognosis…”, which is not suitable, rather prognostic factors. (line 147, 156

Material and method

Line 64 “All included patients had been submitted to total laryngectomy and followed for a minimum of 60 months of follow-up time.”, of follow-up time – is not necessary.

Line 78 - 500 cl is 5 liters, is it possible? Could you precise what alcoholic beverages you consider?

Results

Line 109 “…21 lesions (27.3%) were staged as T2…” Unfortunately this group of patients remains in contradiction to your inclusion criteria of advanced laryngeal cancer. Why T2 patients were treated with total laryngectomy?

Line 114 “ ….whereas eight underwent radiotherapy plus chemotherapy.”  - Could you explain what was the reason for subsequent radio and chemotherapy, because it is not a standard procedure in R0 surgery of laryngeal cancer?

Moreover for comprehensive evaluation it would be advisable to include to the analysis the patients’ comorbidities and the histopathological assessment of the radicality of resection with margin status.

Table 1. There is an error in unit of alcohol use (cc)

Line 126 “ ….10 died after local recurrence,”; from your data, rather after loco-regional recurrence

Line 152  “A significant difference was also found…” - p=0,058 is not significant

Line 163 “21 patients died (seven from T or N recurrence…” rather from loco-regional recurrence

Table 5. Incorrect categories for alcohol use (No/<20cig/day , >20 cig/day)

Discussion

In your discussion you do not confront at all your negative findings regarding other prognostic factors that influence the outcomes in advanced laryngeal cancer: nodal status, tumor size, patients age, that have been confirmed on large study groups, eg.:

J Cancer. 2021; 12(4): 1220–1230. Long-term survival trend after primary total laryngectomy for patients with locally advanced laryngeal carcinoma.

Cancer 2020 Nov 15;126(22):4905-4916.  Association between postoperative complications and long-term oncologic outcomes following total laryngectomy: 10-year experience at MD Anderson Cancer Center

PLoS One. 2017 Jul 14;12(7):e0179371. Impact of stage, management and recurrence on survival rates in laryngeal cancer.

Moreover you admit that the limitation of your study is the number of study group, which in fact is quite small for 10 years (7-8 laryngectomies per year). Could you please give some explanation of such small representation of your study group?

Author Response

ANSWER TO REVIEWER 1

The article investigates the impact of clinical–demographic data on overall survival, disease-free survival and disease-specific survival in patients with advanced-stage laryngeal cancer who underwent total laryngectomy. The manuscript was presented in an intelligible fashion and written in standard English. The weaknesses of the paper include:

Introduction

Line 38 you citated “ ….with the highest number of deaths recorded in Europe [2].” - This data remains in contradiction with WHO International Agency for Research on Cancer information

https://gco.iarc.fr/today/online-analysis table?v=2020&mode=population&mode_population=continents&population=900&populations=900&key=asr&sex=0&cancer=14&type=1&statistic=5&prevalence=0&population_group=0&ages_group%5B%5D=0&ages_group%5B%5D=17&group_cancer=1&include_nmsc=1&include_nmsc_other=1

Answer: Data obtained by the site that you suggest  confirm what reported from the citation [2] The last data in the IARC/WHO database concerning  the deaths in 2015  show a highest number of deaths in Europe (https://www-dep.iarc.fr/WHOdb/table2.asp)

Line 41 you stated “In recent years, the number of open preservation surgeries has increased.” - Could you please give some evidence from the literature to support this statement, because in my opinion the increase is present but in endoscopic intralaryngeal procedures not in open partial laryngectomies.

We precise in the text: “In recent years, the number of open preservation surgeries has increased,open partial laryngectomies have been demonstrated to be a valuable alternative to total laryn-gectomy for  surgical salvage in laryngeal squamous cell carcinoma (LSCC) after chemo-oradiotherapy failure and transoral laser surgery recurrence. (lines 42-44)

Line 50 and thorough other section of the manuscript you use the phrase “….as risk factors of prognosis…”, which is not suitable, rather prognostic factors. (line 147, 156)

 We correct in line 50

Material and method

Line 64 “All included patients had been submitted to total laryngectomy and followed for a minimum of 60 months of follow-up time.”, of follow-up time – is not necessary.

We correct in line 68

Line 78 - 500 cl is 5 liters, is it possible? Could you precise what alcoholic beverages you consider?

 We are sorry for the mistake we considered the unit of mL

Results

 Line 109 “…21 lesions (27.3%) were staged as T2…” Unfortunately this group of patients remains in contradiction to your inclusion criteria of advanced laryngeal cancer. Why T2 patients were treated with total laryngectomy?

As advanced laryngeal cancer is intended stage III-IV as reported in lines 111-112. The T2 laryngeal cancer had a posterior or transglottic extension and no eligible for conservative treatment or partial laryngectomy.

Line 114 “ ….whereas eight underwent radiotherapy plus chemotherapy.”  - Could you explain what was the reason for subsequent radio and chemotherapy, because it is not a standard procedure in R0 surgery of laryngeal cancer?

According to guidelines, tumors with extracapular invasion , more than 1 lymphnode positive and T4 with extralaryngeal diffusion need adjuvant therapy.

Moreover for comprehensive evaluation it would be advisable to include to the analysis the patients’ comorbidities and the histopathological assessment of the radicality of resection with margin status.

All patients had negative surgical margins (line 111). We included the comorbidities  in the results lines 123-124.

Table 1. There is an error in unit of alcohol use (cc)

Thank you, we have corrected.

Line 126 “ ….10 died after local recurrence,”; from your data, rather after loco-regional recurrence

We agree and have changed.

Line 152  “A significant difference was also found…” - p=0,058 is not significant

We intended significant but not statistically significant however we changed the term significant with “evident”.

Line 163 “21 patients died (seven from T or N recurrence…” rather from loco-regional recurrence

We agree and have corrected

Table 5. Incorrect categories for alcohol use (No/<20cig/day , >20 cig/day)

 Thank you, we have corrected.

Discussion

In your discussion you do not confront at all your negative findings regarding other prognostic factors that influence the outcomes in advanced laryngeal cancer: nodal status, tumor size, patients age, that have been confirmed on large study groups, eg.:

J Cancer. 2021; 12(4): 1220–1230. Long-term survival trend after primary total laryngectomy for patients with locally advanced laryngeal carcinoma.

This study doen’t take into consideration smoking habit and alchol consuming.

Cancer 2020 Nov 15;126(22):4905-4916.  Association between postoperative complications and long-term oncologic outcomes following total laryngectomy: 10-year experience at MD Anderson Cancer Center

PLoS One. 2017 Jul 14;12(7):e0179371. Impact of stage, management and recurrence on survival rates in laryngeal cancer.

The authors in this study take into consideration the impact of the postoperative complications on the survival , there are no data about smoke and alchol habits.

Moreover you admit that the limitation of your study is the number of study group, which in fact is quite small for 10 years (7-8 laryngectomies per year). Could you please give some explanation of such small representation of your study group?

This regards laryngeal cancer treated by chemoradiotherapy and surgery , the patients trated by total laryngectomy amount to 11.1%  of the total including early and advanced stage od disease.

We retain that these studies cannot compared to our one.

Moreover you admit that the limitation of your study is the number of study group, which in fact is quite small for 10 years (7-8 laryngectomies per year). Could you please give some explanation of such small representation of your study group?

We excluded from the study patients lost at follow-up or with uncomplete data, furtherome pur hospital serve a population of 1 milions of  inhabitants.

Reviewer 2 Report

Thank you for your submission.  This is a well organized study with clear results and findings indicating a link between smoking and prognosis for patients with advanced laryngeal scc.  

Please clarify: 

  • All patients underwent primary laryngectomy (none in the salvage setting)
  • was the smoking variable collected at the time of diagnosis (never, <20 cig/day or >20 cig/day)
  • Why not also report smoking pack-years?  Given the ongoing damaging effects of past smoking, it is important to report the smoking variable in terms of pack-years as well.
  • Did all patients have squamous cell carcinoma (SCC)? please clarify.  The term "laryngeal carcinoma" is used.
  • Other variables that should be reported in your analysis should include peri-neural invasion, margins, lymphovascular invasion, extranodal extension, presence of worst pattern of invasion 5, etc.  This type of study is incomplete without these variables.  
  • Additionally, please acknowledge that occasionally laryngectomy patients continue to smoke post-op through their stoma.  

Author Response

ANSWERS TO REVIEWER 2

  • All patients underwent primary laryngectomy (none in the salvage setting)

No were excluded from the study as reported in the methods.

  • was the smoking variable collected at the time of diagnosis (never, <20 cig/day or >20 cig/day)

Yes, it was.

  • Why not also report smoking pack-years?  Given the ongoing damaging effects of past smoking, it is important to report the smoking variable in terms of pack-years as well.

We prefer to collect data about cigarette smoked per day because generally patients didn’nt know to answer about how many pack-year.

  • Did all patients have squamous cell carcinoma (SCC)? please clarify.  The term "laryngeal carcinoma" is used.

Yes , all were squamous cell carcinoma, as already reported in the line 59

  • Other variables that should be reported in your analysis should include peri-neural invasion, margins, lymphovascular invasion, extranodal extension, presence of worst pattern of invasion 5, etc.  This type of study is incomplete without these variables.  

This is a retrospecive study we have data on resection margins and extranodal extension. We  reported in the text these informations at line 111 and lines 119-120.

  • Additionally, please acknowledge that occasionally laryngectomy patients continue to smoke post-op through their stoma.  

Yes, I know and I saw  patients to smoke through the stoma and generally they stopped after some months after the operation.

Round 2

Reviewer 2 Report

I have not seen enough revisions made to the original manuscript.  Smoking should be reported as pack years, as this is the most common and widely used method of reporting smoking history.  I have not seen the addition of other important pathologic variables such as PNI, LVI and ENE.  Additionally, given that we now use the 8th version of the AJCC staging system, this staging system should be used in this manuscript.  When looking at advanced laryngeal squamous cell carcinoma requiring a total laryngectomy as a primary treatment modality, you should avoid looking at overall stage III-IV (which will include T2 disease) and focus on T3-4.  Patients with T2 should rarely be offered a TL as a primary treatment modality - better suited for C/XRT or conservational surgery (open or endoscopic).  

Author Response

Answers To the Reviewer 2 comments

  1. Smoking should be reported as pack years, as this is the most common and widely used method of reporting smoking history. 

To evaluate smoking habit, pack/year is not the only one method used in the literature. This method is mostly used in the USA. Many  authors use cigarette/day , we published previous paper in which we used the same parameter to evaluate smoking habit. Furthermore many authors report the smoking use as never, ex-smokers and smokers. We report one of our previous article in which we report the smoke habit as cigarettes/day.

Allegra E, Trapasso S, La Boria A, Aragona T, Pisani D, Belfiore A, Garozzo A. Prognostic role of salivary CD44sol levels in the follow-up of laryngeal carcinomas. J Oral Pathol Med. 2014 Apr;43(4):276-81. doi: 10.1111/jop.12129. PMID: 24822267.

  1. I have not seen the addition of other important pathologic variables such as PNI, LVI and ENE.

This is a retrospective study we have data on resection margins and extranodal extension,  data on PNI and LVI were not collected, while we reported in the text  information about resection margins and ENE at line 111 and lines 119-120.

  1. Additionally, given that we now use the 8th version of the AJCC staging system, this staging system should be used in this manuscript.

The AJCC staging system reported is that used at the time of the diagnosis, We don’t understand why we have to use  actual staging system for retrospective analysis.

  1. When looking at advanced laryngeal squamous cell carcinoma requiring a total laryngectomy as a primary treatment modality, you should avoid looking at overall stage III-IV (which will include T2 disease) and focus on T3-4.  Patients with T2 should rarely be offered a TL as a primary treatment modality - better suited for C/XRT or conservational surgery (open or endoscopic).  

We confirm that patients T2  treated were not eligible for conservative surgical treatment. They were T2N+ then they have been included in the advanced laryngeal cancer. If they were candidate for conservative surgery, we were happy for the patients and we had not problem to treat the patient with transoral laser surgery or by open preservation surgery. We report some of the article published from our team about conservative laryngeal surgery.

Allegra E, Bianco MR, Mignogna C, Drago GD, Modica DM, Puzzo L. Early Glottic Cancer Treated by Transoral Laser Surgery Using Toluidine Blue for the Definition of the Surgical Margins: A Pilot Study. Medicina (Kaunas). 2020 Jul 3;56(7):334. doi: 10.3390/medicina56070334. PMID: 32635245; PMCID: PMC7404472.

Allegra E, Saita V, Azzolina A, De Natale M, Bianco MR, Modica DM, Garozzo A. Impact of the anterior commissure involvement on the survival of early glottic cancer treated with cricohyoidoepiglottopexy: a retrospective study. Cancer Manag Res. 2018 Nov 8;10:5553-5558. doi: 10.2147/CMAR.S182854. PMID: 30519103; PMCID: PMC6234988.

Allegra E, Franco T, Trapasso S, Domanico R, La Boria A, Garozzo A. Modified supracricoid laryngectomy: oncological and functional outcomes in the elderly. Clin Interv Aging. 2012;7:475-80. doi: 10.2147/CIA.S38410. Epub 2012 Nov 8. PMID: 23152678; PMCID: PMC3496192.

Allegra E, Lombardo N, La Boria A, Rotundo G, Bianco MR, Barrera T, Cuccunato M, Garozzo A. Quality of voice evaluation in patients treated by supracricoid laryngectomy and modified supracricoid laryngectomy. Otolaryngol Head Neck Surg. 2011 Nov;145(5):789-95. doi: 10.1177/0194599811416438. Epub 2011 Jul 26. PMID: 21791705.

De Seta D, Campo F, D'Aguanno V, Ralli M, Greco A, Russo FY, de Vincentiis M. Transoral laser microsurgery for Tis, T1, and T2 glottic carcinoma: 5-year follow-up. Lasers Med Sci. 2020 May 29. doi: 10.1007/s10103-020-03049-4. Epub ahead of print. PMID: 32472425.

Campo F, Zocchi J, Ralli M, De Seta D, Russo FY, Angeletti D, Minni A, Greco A, Pellini R, de Vincentiis M. Laser Microsurgery Versus Radiotherapy Versus Open Partial Laryngectomy for T2 Laryngeal Carcinoma: A Systematic Review of Oncological Outcomes. Ear Nose Throat J. 2021 Feb;100(1_suppl):51S-58S. doi: 10.1177/0145561320928198. Epub 2020 Jun 8. PMID: 32511005.

De Virgilio A, Fusconi M, Gallo A, Greco A, Kim SH, Conte M, Alessi S, Tombolini M, de Vincentiis M. The oncologic radicality of supracricoid partial laryngectomy with cricohyoidopexy in the treatment of advanced N0-N1 laryngeal squamous cell carcinoma. Laryngoscope. 2012 Apr;122(4):826-33. doi: 10.1002/lary.23178. Epub 2012 Feb 16. PMID: 22344785.

Bussu F, Paludetti G, Almadori G, De Virgilio A, Galli J, Miccichè F, Tombolini M, Rizzo D, Gallo A, Giglia V, Greco A, Valentini V, De Vincentiis M. Comparison of total laryngectomy with surgical (cricohyoidopexy) and nonsurgical organ-preservation modalities in advanced laryngeal squamous cell carcinomas: A multicenter retrospective analysis. Head Neck. 2013 Apr;35(4):554-61. doi: 10.1002/hed.22994. Epub 2012 Apr 12. PMID: 22495830.